# Versatile stochastic dot product circuits based on nonvolatile memories for high performance neurocomputing and neurooptimization

M.R. Mahmoodi [1], M. Prezioso[1] & D.B. Strukov[1]*

The key operation in stochastic neural networks, which have become the state-of-the-art approach for solving problems in machine learning, information theory, and statistics, is a stochastic dot-product. While there have been many demonstrations of dot-product circuits and, separately, of stochastic neurons, the efficient hardware implementation combining both functionalities is still missing. Here we report compact, fast, energy-efficient, and scalable stochastic dot-product circuits based on either passively integrated metal-oxide memristors or embedded floating-gate memories. The circuit's high performance is due to mixed-signal implementation, while the efficient stochastic operation is achieved by utilizing circuit's noise, intrinsic and/or extrinsic to the memory cell array. The dynamic scaling of weights, enabled by analog memory devices, allows for efficient realization of different annealing approaches to improve functionality. The proposed approach is experimentally verified for two representative applications, namely by implementing neural network for solving a four-node graph-partitioning problem, and a Boltzmann machine with 10-input and 8-hidden neurons.

[1] Electrical and Computer Engineering Department, University of California Santa Barbara, Santa Barbara, CA 93117, USA. *email: strukov@ece.ucsb.edu

Computations performed by the brain are inherently stochastic.[1–7] At the molecular level, this is due to stochastic gating of ion channels of the neurons[3] and probabilistic nature of transmitter release at the synaptic clefts.[4] Noisy, unreliable molecular mechanisms are the reason for getting substantially different neural responses to the repeatable presentations of identical stimuli, which, in turn, allows for a complex stochastic behavior, such as Poisson spiking dynamics.[2,5,6] Though noise is always detrimental for conventional digital circuits, a very low signal-to-noise ratio (SNR) of neuronal signals[7] has been suggested to play an important role in the brain functionality, e.g., in its ability to adapt to changing environment[1,2,5,6], as well as for achieving low energy operation[8].

It is therefore not surprising that many developed artificial neural networks rely on stochastic operation. For instance, probabilistic synapses could be used as a main source of randomness for reinforcement learning[9], or as regularizers during training, significantly improving classification performance in spiking neural network[10]. In such networks, probabilistic synapses also allow relaxing the requirements for synaptic weight precision due to temporal averaging over a spike train[11].

One of the prominent examples is a Boltzmann machine, a recurrent stochastic neural network with bidirectional connections[12,13], which can be viewed as a generalization of the Hopfield network[14,15]. In its simplest form, the Boltzmann machine is a network of $N$ stochastic binary neurons. At each discrete-time instance, the network is in a certain state, which is characterized by binary $V_i$ outputs of its neurons. The network dynamics is arranged to model thermal equilibrium, at certain temperature $T$, of a physical system with energy $E$:

$$E = -\sum_{i=1}^{N} V_i I_i \quad I_i = \sum_{j=1}^{N} G_{ij} V_j + I_i^{\mathrm{b}}, \quad (1)$$

where $I_i$ and $I_i^{\mathrm{b}}$ are analog input and bias, which are typically represented by currents in the circuit implementation, while $G_{ij}$ is a synaptic weight (conductance) between $i$th and $j$th neurons. The network state is updated by changing the state of the randomly chosen neurons. The probability of a neuron being switched to the digital state "1" with amplitude $V_{\mathrm{ON}}$—in other words, turned "on" or being activated—is a sigmoid function of its input i.e.,

$$\Pr(V = V_{\mathrm{ON}}) = \frac{1}{1 + \exp(-I'/T)}. \quad (2)$$

Here $T$ is a dimensionless temperature, and $I'$ is a normalized input current $I' = I/I_{\max}$, where $I_{\max}$ is the largest possible neuron input current, common for the whole network. The process of simulated annealing, in which initially high temperature is gradually decreased over time, helps the network to escape local energy minima[15].

As a stochastic version of Hopfield networks, the Boltzmann machine, combined with simulated annealing, is a powerful approach for solving combinatorial optimization problems[15]. Moreover, such networks can be utilized to compute conditional and marginal probabilities by fixing the states of some neurons and sampling outputs of the unclamped ones. Such functionality is central for many Boltzmann machine derivatives, such as deep-belief networks[16], and Bayesian model computing[17]. The invention of the restricted Boltzmann machine (RBM)[12,18] and efficient training algorithms[19] have led to its widespread use in machine learning tasks[18], including dimensionality reduction[20], classification[21], and notably, collaborative filtering, for example enabling the best performance in the Netflix movie prediction challenge[22]. Furthermore, owing similarities to Markov random fields, Boltzmann machines have found many applications in statistics and information theory[18].

The stochastic dot-product computation described by Eqs. 1, 2 is the most common operation performed during inference and training in Boltzmann machines and its variants, and hence its efficient hardware realization is of utmost importance. By now, there have been many demonstrations of high performance dot-product circuits, most importantly including analog and mixed-signal implementations based on metal-oxide memristors[23–27], and phase-change[28,29] and floating-gate memories[30], developed in the context of neuromorphic inference applications[31–33]. Analog dot-product circuits based on metal-oxide memristors have been also demonstration in the context of neural optimization[34,35].

For Boltzmann machines, the stochastic functionality can be realized in neural cells, peripheral to the array of memory cells, rather than at much more numerous synapse locations, which somewhat relaxes the design requirements. Still, prior studies showed that even with a relatively large synapse to neuron ratio (~1000) and deterministic dot-product functionality, the neuron circuitry may constitute a substantial part of the neuromorphic inference systems[30,36,37]. Because of such concerns, purely CMOS implementations, see, e.g., CMOS probabilistic gates[38] and CMOS-based Ising chip for combinatorial optimization problems[39], may not be very appealing. (These challenges are somewhat similar to CMOS-implemented random number generators in the context of hardware security applications, and served as a motivation to use emerging memory device technologies[40,41].)

The implementation overhead of stochastic functionality might be less of a problem for some memory devices, in which switching between memory states is inherently stochastic, e.g., ferromagnetic[42,43], phase-change[44,45], ionic[46,47] and thermally driven metal-oxide[48], and solid-state electrolyte devices[49,50]. Unfortunately, many of such devices come with other severe challenges. For instance, an efficient implementation of the large-scale dot-product computation is a major challenge for magnetic devices. The hybrid option of combining magnetic stochastic neurons with the already mentioned mixed-signal dot-products is not appealing, because an extreme energy efficiency of spin-based computing is typically compromised by the interface with charge-based devices. The technology of magnetic devices is also quite immature judging by low-complexity of experimental demonstrations[51,52] (Supplementary Table 1). The biggest challenges for the remaining devices are low switching endurance and variations in switching characteristics.

In this paper, we propose to utilize intrinsic and extrinsic current fluctuations in mixed-signal circuits based on analog-grade nonvolatile memories to implement scalable, versatile, and efficient stochastic dot computation. The deterministic version of such dot-product circuits have been extensively investigated due to their potentials for high speed, high density, and extreme energy efficiency—see, e.g., refs. [53,54]. Unlike many prior proposals[42,43,51,52], our approach is suitable for large-scale dot-product circuits and has no endurance restrictions for inference operation, which is typical for other proposals[44–50,55–57]. We experimentally verify stochastic dot-product circuits based on metal-oxide memristors and embedded floating-gate memories by implementing and testing Boltzmann machine networks with non-binary weights and hardware-injected noise. We further demonstrate how scaling of synaptic weights during operation can be used for a very efficient annealing implementation to improve functional performance.

## Results

**Stochastic dot-product circuit.** Figure 1a shows the investigated current-mode analog circuit based on nonvolatile memories, in

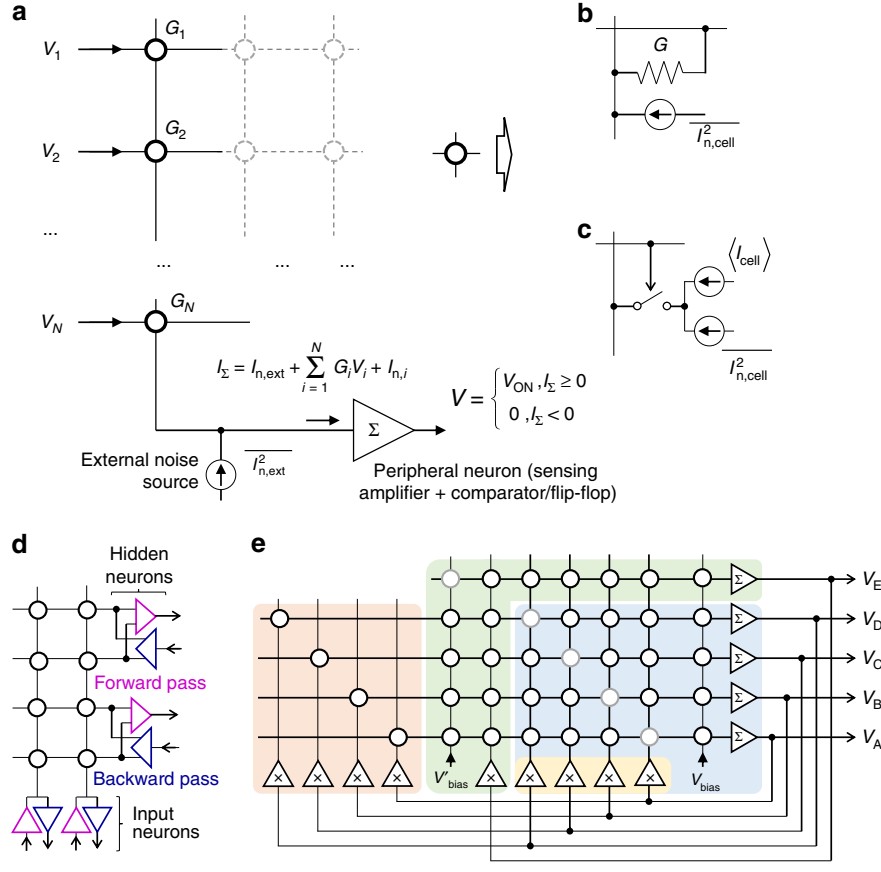

**Fig. 1** Stochastic dot-product circuit and its applications in neurocomputing. **a** Circuit schematics for the design with current-mode sensing with crosspoint device implementation based on (**b**) memristors and (**c**) floating-gate memories. The equations in figure corresponds to memristor implementation, while their modified version for floating-gate design are described in text. **d** An example of the considered differential-pair Boltzmann machine implementation. **e** The implementation of generalized Hopfield neural network. The blue background highlights the baseline implementation. The yellow, green, and red backgrounds highlight additional circuitry for the proposed "stochastic", "adjustable", and "chaotic" approaches, respectively. The gray shaded circles show synaptic weights which are typically set to zero. Labels "Σ"/ "x" inside amplifier symbols denote summation / scaling. For clarity, panel a does not show bias currents, panels (**a**) and (**e**) show single-ended network, while panel (**d**) shows (**a**) small two-input, two-hidden neurons fragment of the considered network

which vector-by-matrix operation is efficiently implemented on the physical level due to Ohm's and Kirchhoff's laws. For memristor-based circuits (Fig. 1b), the weights are encoded with device conductances, so that the current flowing into the virtually grounded neuron is given by $\Sigma_i G_i V_i$ and the network operation is described by equations in Fig. 1a. For the considered discrete-state networks, a crosspoint floating-gate transistor can be conveniently viewed as a switch connecting a current source to a neuron's input (Fig. 1c). The cell currents $I_{cell}$ at voltage bias $V_{ON}$ used at network operation are pre-set according to the desired synaptic weight. The neuron input current is given by $\Sigma_i I_{cell,i}(V_i)$, where $I_{cell}$ $(V_{OFF}) \approx 0$ when digital "0" is applied to the cell's switch.

The circuit noise is detrimental to the deterministic dot-product operation and, e.g., defines the lower bound on the memory cell currents for a desired computing precision[58]. The main difference with prior work is that in the proposed operation the noise is exploited for stochastic functionality. Specifically, two types of noise sources are considered: intrinsic noise to each memory cell and externally added noise to each output, e.g., from additional fixed-biased memory cells or using the input-referred current noise of peripheral circuits.

To analyze stochastic operation, let us consider normally distributed independent noise sources. This assumption is justified due to the dominant white (thermal and/or shot)

intrinsic noise for the most practical >100 MHz bandwidth operation, which would be realistic for both floating-gate transistor and memristor-based analog circuits in which memory arrays are tightly integrated with peripheral circuits[53,54]. The current $I$ is sampled and latched at the peripheral neuron, which consists of a current-mode sensing circuitry feeding, in the case of a discrete-time networks, a digital flip flop (Fig. 1a). The flip flop effectively implements a step function of the sampled value, so that the probability of latching a digital "1" state is

$$\Pr(V = V_{ON}) = \frac{1}{2} + \frac{1}{2}\,\mathrm{erf}\,\frac{I}{\sqrt{2}\sigma}, \tag{3}$$

where σ is the standard deviation of the output current.

There are two characteristic regimes for stochastic operation defined by Eq. 3. If thermal noise dominates, the fluctuations of an output current would be independent of its average value. In this case, Eq. 3 matches almost exactly the sigmoid function of Eq. 2 given that temperature is inversely proportional to a peak SNR $I_{max}/\sigma$ as

$$T = \frac{\sqrt{2\pi}\sigma}{4I_{max}}. \tag{4}$$

With predominant shot noise behavior, $\sigma \propto \sqrt{I}$. Even in this case, Eq. 3 could closely approximate Eq. 2 assuming some

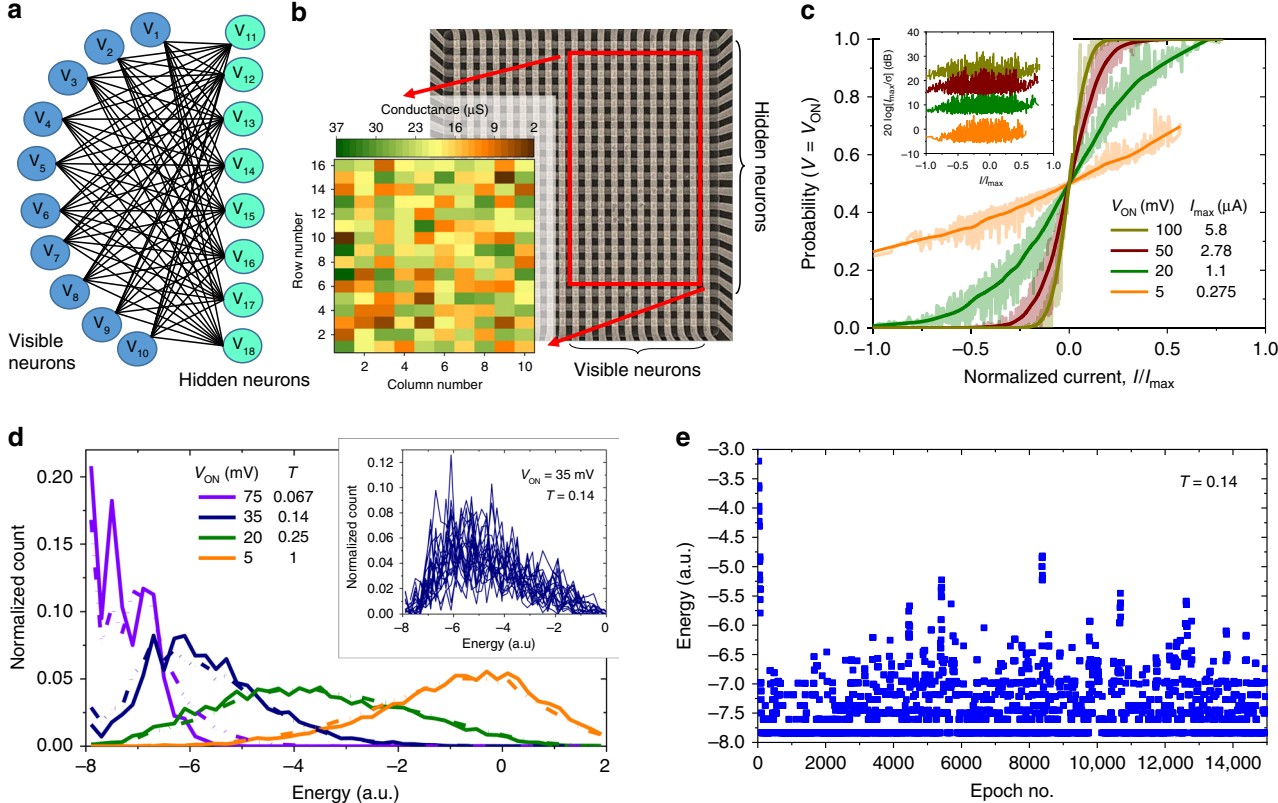

**Fig. 2** Memristor-based restricted Boltzmann machine. **a** A bipartite graph of the considered RBM network and (**b**) its implementation with passively integrated metal-oxide memristors. The red rectangle highlights the utilized area of 20 × 20 crossbar array, while the inset shows the conductance map, measured at 50 mV, after programming devices to the desired states. Note that though the neurons in Boltzmann machine are typically partitioned into visible and hidden ones, for simplicity, we use the same notations for both types. **c** Measured stochastic neuron transfer functions at several $V_{ON}$, i.e., different effective computing temperatures, for the hidden neuron #2 of the implemented network, which is attached to rows #3 and #4 of the crossbar circuit. The inset shows peak signal-to-noise ratios across full range of neuron's input currents. $I_{max}$ corresponds to the largest input current to the neuron #2. Some minor, unwanted SNR dependence on the input current is due to the artifacts of the experimental setup. **d** Measured (solid) and modeled (dash-dot) energy distributions. The inset shows measured energy distributions for the specific temperature, collected over 20 different 1000-epoch spans. **e** An example of the simulated evolution of energy for the specific temperature. All neurons are initialized to zero state at the beginning of this simulation. In all experiments, the neuron's input currents were sampled at 1 MHz bandwidth, while integrating noise above this frequency. On panels (**d**) and (**e**), the temperature is computed relative to the largest possible current, which corresponds to all ten differential synaptic weights set to the maximum value of 32 μS

effective temperature—see Supplementary Fig. 1 and its discussion in Supplementary Note 1 for more details.

The first regime is representative of intrinsic thermal noise in metal-oxide memristors. Indeed, shot noise in such devices would be negligible due to typically diffusive electron transport[59] and relatively small $V_{ON}$, which should be not much larger than a thermal voltage at room temperature to avoid disturbance of memory state. Note that intrinsic thermal noise is independent of the applied voltage and will be contributed by all memristors in the column, even zero-biased crosspoint devices, thus excluding any input dependence (Fig. 1b). On the other hand, the intrinsic shot noise is characteristic of a ballistic transport in nanoscale floating-gate transistors with sub-10-nm channels[60,61]. This noise can be completely cut off by opening the cell's switch (Fig. 1c). For both implementations, the effective computing temperature can be dynamically varied by changing $I_{max}$. Moreover, the scaling constant can be uniquely selected for each array's input by adjusting its voltage amplitudes—see, e.g., amplifiers marked with "x" in Fig. 1e.

Stochastic dot-product operation and runtime temperature scaling are demonstrated next in the context of two applications.

**Memristor-based RBM**. In our first experiment, we focused on the demonstration of an RBM using 20 × 20 crossbar circuits with passively integrated $Pt/Al_2O_3/TiO_{2-x}/Pt$ memristors (Fig. 2), fabricated using the device technology reported in ref. [24]. The integrated memristors are sufficiently uniform for programming with less than 5% tuning error, and have negligible conductance drift over time. Limiting the applied voltage bias across memristors to $| V_{ON} | \leq 100$ mV prevents disturbance of memory states during the network operation. At such small voltages, the memristor I–V characteristics are fairly linear, with $I(V_{ON})/(2I(V_{ON}/2)) < 1.02$ for all conductive states[24]. (More details on the memristor technology and crossbar circuit operation is provided in Methods section.)

The studied bidirectional network consists of 10 visible and 8 hidden neurons (Fig. 2a) with synaptic weights implemented as differential memristor pairs. Each visible neuron is connected to a single vertical electrode of the crossbar, while each differential hidden neuron is attached to two horizontal crossbar electrodes (Fig. 1d). The forward propagation of the information, i.e., from visible neurons to hidden ones, and differential sensing is performed similarly to previous work[24]. In the backward pass, digital "1" input from the hidden neuron is implemented by applying $\pm V_{ON}$ to the corresponding differential pair of lines, while grounding both lines for zero input. The current is then sensed at single-line input of the visible neuron.

For simplicity, we study the network with zero bias weights. The remaining weights were chosen by first generating random real numbers within $[-1, +1]$ range. These values were mapped to $-32\,\mu S$ to $+32\,\mu S$ at 50 mV maximum conductance range of a differential pair using the $20\,\mu S$ conductance bias and the $4–36\,\mu S$ dynamic range of individual devices. After such individual device conductances had been determined, memristors were programmed using automated tuning algorithms[62] with the 5% tuning accuracy to the desired states (inset of Fig. 2b).

Figure 2c shows the stochastic dot-product results when utilizing external noise, which was injected directly in the hardware from the read-out circuitry. The noise spectrum is flat at $>\sim$1 KHz frequencies (Supplementary Fig. 2a), which results in approximately fixed standard deviation of the injected noise (inset of Fig. 2c) for the studied 1 MHz bandwidth. Specifically, these results were obtained by applying all possible digital inputs to the hidden neuron #2, and collecting 100,000 samples of the crossbar array output currents for each specific input, while emulating the peripheral circuitry in software. (A possible implementation of peripheral circuits is shown in Supplementary Fig. 3.) The effective computing temperature, i.e., the slope of sigmoid function, is controlled by $V_{ON}$.

In our main RMB experiment, we first apply randomly generated digital voltages to the vertical crossbar lines connected to visible neurons, then sample output currents on the horizontal crossbar lines feeding hidden neurons, and convert sampled values to the new digital voltages of hidden neurons according to the signs of the corresponding differential currents. Note that only functionality of a sensing circuit and latch (i.e. applying step function to the sensed currents and holding the resulting digital value) are realized in a software, while the probability function of Eq. 3 is implemented directly in the hardware. In the next step, the calculated voltages at the hidden neurons are applied to horizontal lines and the new voltages at the input neurons are computed similarly to the forward pass. The voltages at the input and hidden neurons represents the new state of the network after one forward/backward state update ("epoch") and are used to calculate its energy according to Eq. 1. These updates are repeated multiple times in a single run of the experiment.

Figure 2d shows the results of such experiment, namely the energy distributions at different effective computing temperatures calculated from Eq. 4. Each distribution corresponds to the measured energies in the final 500 epochs of a single run (see an example for such run in Fig. 2e), averaged across 100 different trials, that start with randomly chosen initial neuron states. For a wide range of effective computing temperatures, the experimentally measured data are in good agreement with simulations, which were based on the stochastic binary neuron with ideal sigmoid probability function. Note that because of bipartite network topology, the system quickly converges to thermal equilibrium, which is indirectly confirmed by comparing energy distributions over different time periods (inset of Fig. 2d).

**Neurooptimization based on floating-gate memories**. In our second experiment, we investigated implementation of generalized Hopfield network with embedded NOR flash memory for solving combinatorial optimization problem (Fig. 3). The experimental work was performed on $6 \times 10$ integrated array of supercells (Fig. 3a), using 55-nm technology modified from the commercial process for analog tuning[63]. One supercell (Fig. 3a) hosts two floating-gate transistors sharing a common source terminal, so that there are 120 memory cells in such array. The subthreshold currents of crosspoint transistors can be tuned uniquely and precisely for each cell within a very wide dynamic range by adjusting the charges at the floating gates[63], enabling

very efficient implementation of dot-product operation in which inputs are applied to word gate (WG) lines and output currents are sensed from the drain (D) lines[46–49,54]. Furthermore, the currents can be simultaneously scaled (and even completely suppressed), without re-tuning, for all cells sharing the same coupling gate (CG)/WG line, by controlling CG and/or WG voltage amplitudes, while keeping other cell's terminals biased under typical reading conditions. More details on the utilized embedded NOR flash memory devices and circuits are provided in Method section.

Figure 3b shows the results of stochastic dot-product operation for the flash memory implementation. For these measurements, currents of 10 cells, sharing a drain line of the memory array, were set with 20% tuning precision to 175 nA, which is representative value for the considered experiment. After that, 20,000 samples of single-ended bit-line currents were collected at 10 KHz bandwidth for 30 randomly selected inputs. Similar to RBM study, fixed white noise was added externally directly from the read-out circuit (inset of Fig. 3b, Supplementary Fig. 2b), while other peripheral functions were emulated in the software. To consider different neuron's input currents, $m$ cells (out of 10 total) on the bit line were randomly chosen, i.e., a specific voltage was applied to the selected $m$ WG lines, while the remaining cells were disabled by grounding their WG lines. This experiment was repeated three times for each $m$ from 1 to 10. The effective computing temperature was controlled by adjusting CG voltage.

Our specific focus is on solving graph-partitioning problem with parameters specified in Fig. 3c. Supplementary Note 2 provides more details on the problem formulation and its neural network implementation. The neural network weights were mapped to the cell currents using $(I_{cell})_{max} = 1.0\,\mu A$ (Supplementary Eq. 6), which resulted in comparable to memristor study range of SNRs. To demonstrate versatility of the proposed circuits, four different variations of Hopfield networks were considered for solving this combinatorial optimization problem: an original approach (labeled as "baseline")[14]; a scheme with dynamically adjusted problem/energy function ("adjustable"); a network with chaotic annealing ("chaotic")[64]; and, finally, a generalized Hopfield network with simulated annealing implementation, which is enabled by stochastic dot-product circuits ("stochastic"). For all approaches, the implemented network is discrete time/state with state updates performed sequentially for a randomly selected neurons during operation. The array and bias conductances (Fig. 3d) were calculated according to Supplementary Eq. 6.

More specifically, the proposed "adjustable" approach draws inspiration from the work on quantum annealing[65], in which an initial problem is modified to ease convergence to a global optimum. Similarly, we modified the problem by adding an additional node with relatively large weight and zero-weight edges (Fig. 3c). The additional node weight was exponentially decreased from 50 to 0.2 at each update, thus gradually turning the mapped problem and its energy function (Supplementary Eq. 4) to those of the original one. In the hardware implementation, the extra node was realized by extending the original memory cell array by one column and one row (highlighted with green background in Fig. 1e) and decreasing WG voltage from 1.2–0.2 V. Note that the additional bias line was required to separate contribution of bias currents from the original node weights and that of the added one (Supplementary Eq. 5).

For the chaotic annealing approach, we followed the idea of ref. [64] to utilize transient chaos for better convergence. The chaotic behavior was facilitated by initially employing large negative diagonal synaptic weights ($I_{cell} = -1.2\,\mu A$ at $V_{CG} = 1.5$ V and $V_{WG} = 1.2$ V) which were encoded in a separate array of cells (Fig. 1e). These weights were decreased linearly to ~0 with

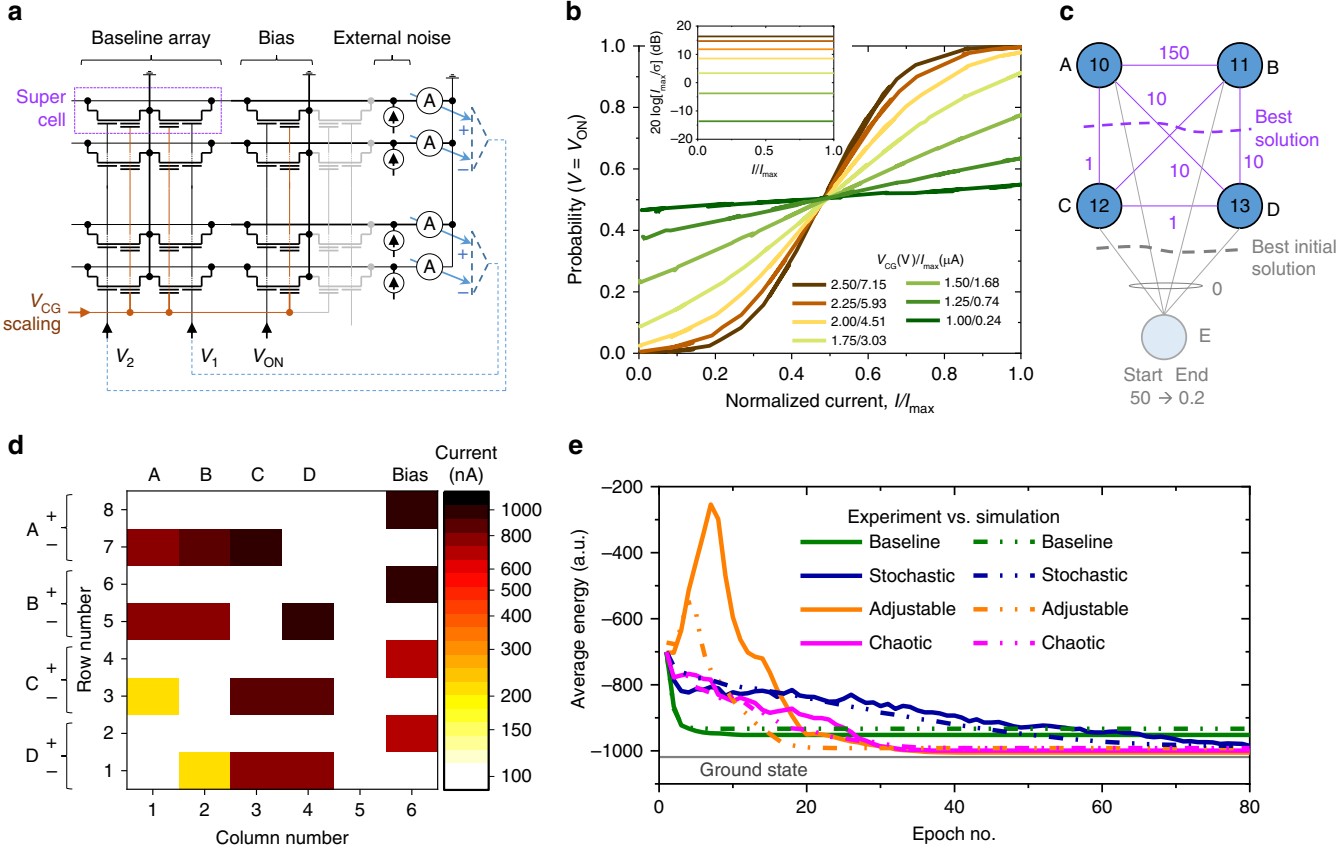

**Fig. 3** Neurooptimization based on floating-gate memory arrays. **a** Schematics of the experiment, shown for clarity for the 2 × 2 supercell array implementing a baseline/stochastic network with two neurons. The parts shown with dashed lines were emulated in software. **b** Measured probability functions for single-ended stochastic neuron at various CG voltages (or effective computing temperatures). The inset shows measured peak SNR across the full range of neuron's input currents. $I_{max}$ is the largest measured input current to the neuron. **c** Implemented graph-partitioning problem with considered edge and node weights. The wavy lines show the cuts corresponding to the best solution. Shaded nodes/edges are used for the method with dynamically adjusted energy function. **d** Conductance map for the main section (highlighted with blue background in Fig. 1e) of the weights in the experiment, after tuning with 3% precision. For the fixed synaptic weights, the tuning was performed at the operating biases. For the variable weights, the tuning was performed at the lowest (largest) CG and WG voltages during operation for the stochastic (adjustable and chaotic) approaches. **e** Simulation and experimental results for neurooptimization. For the first three (chaotic) cases, the data are averaged over 100 (20) runs for each of the initial state, with total of 1600 (320) runs. The energy function for "adjustable" case is calculated by taking into account only four original nodes. The biasing conditions during operation are summarized in Supplementary Table 2

each update by changing WG voltage on these additional cells during runtime from 1.2 to 0 V.

In the baseline, adjustable, and chaotic annealing experiments, all updates were deterministic (i.e. with larger SNR for neuron input currents) due to using larger CG voltages (Supplementary Table 2), and also very low (~20 Hz) operational bandwidth, which further reduced noise impact. For stochastic Hopfield network, the nodes were updated probabilistically at 20 KHz bandwidth according to Eq. (2). To implement simulated annealing, CG voltage was exponentially increased from 1 to 2 V in 80 steps, which corresponds to 80 × decrease in effective computing temperature.

Figure 3e shows the main results of neurooptimization study. The convergence for the baseline approach is fast (see also additional experimental results in Supplementary Fig. 4), though the network often gets stuck in the local minima. As a result, the final energy, averaged over many runs, is substantially higher than the global optimum ("ground state" line in Fig. 3e), which corresponds to the solution shown with a dashed red line in Fig. 3c. On the other hand, optimal solution was almost always found using three remaining approaches. For the adjustable approach, the initial increase in energy of the original 4-node

problem is expected, given the quick convergence to the global energy optimum of the 5-node problem. As the additional node is gradually eliminated from the network, 4-node problem energy quickly drops to below baseline level, resulting in a better solution. This is likely due to the network state being very close to its global minima during convergence and/or more optimal initial state corresponding to the optimal solution of the 5-node problem.

For all considered approaches, experimental data follow very closely simulation results (Fig. 3e). Furthermore, the SPICE simulations at 100 MHz operation bandwidth also show similar performance when using only intrinsic cell noise (Supplementary Fig. 4a).

## Discussion

The considered case studies allow contrasting stochastic dot-product circuit implementations with two representative memory technologies.

The main advantage of floating-gate memory devices is their mature fabrication technology, which can be readily used for implementing practically useful, larger-scale circuits. Their

substantial drawbacks for the considered applications include unipolar electron transport, which, e.g., necessitates using two different sets of cells with similarly tuned conductances, for forward and backward computations in RBM networks. Floating-gate memory cells are also sparser and less scalable, though these deficiencies are somewhat compensated by lower peripheral overhead due to the cells' high input and output impedances[30,54], and also by having more design options in scaling cell currents due to multi-terminal cell structure, which is important for the considered annealing approaches.

Furthermore, there are two specific problems for floating-gate implementation which may lead to a "smearing" of neuron's transfer function. First, for differential current sensing, the total injected (shot) noise depends on the absolute currents at the differential lines, rather than their subtracted value. The problem can be better illustrated by considering two extreme cases, namely when subtracting two smaller similar currents and two larger similar currents on the differential lines. The total differential current could be comparable, though due to the dependence of the intrinsic shot noise on the cell currents, the SNR would be larger (and hence effective temperature smaller) for the latter case. To investigate this issue further, we considered a 100-node graph-partitioning problem with randomly distributed weights and edges within [0,1] interval. Figure 4a shows the corresponding neural network weight map. We then simulated stochastic neuron's transfer functions by adding shot-like noise $\sigma^2 = \alpha I$ to differential lines and considering different combinations of

input currents for all neurons (Fig. 4b). Second, due to variations in subthreshold slopes of floating-gate transistors, there are noticeable changes in relative weights when scaling currents. Specifically, in the ideal case, the relative cell currents (and hence the synaptic weights) should scale similarly when changing CG voltage in the proposed annealing schemes. In practice, however, the cells' currents scale differently due to process-induced variations and voltage dependency of the subthreshold slope. To quantify this issue, we have measured subthreshold characteristics of the 100 devices, which were tuned randomly at $V_{CG} = 2$ V, $V_{WG} = 1.2$ V) to currents ranging from 40 nA to 1 μA (Fig. 4c). Fortunately, extensive modeling results show that the resulting smearing of the stochastic neuron transfer function due to both issues is rather negligible, more so at lower temperatures (Fig. 4d).

On the other hand, metal-oxide memristors are arguably the most prospective candidate for the proposed circuits. Because of input-independent intrinsic noise and linear static $I$–$V$ characteristics at small biases, the non-idealities discussed in Fig. 4 are much less of a problem for memristors. Their major challenge, however, is immature technology requiring substantial improvements in device yield and $I$–$V$ uniformity. The improved device technology should also feature lower cell currents, by approximately two orders of magnitude, to improve system level performance and to allow for high effective temperatures during operation when relying on intrinsic noise in the stochastic dot-product circuits. Due to the linear dependence of the off-state current on the device footprint, cell currents in the utilized

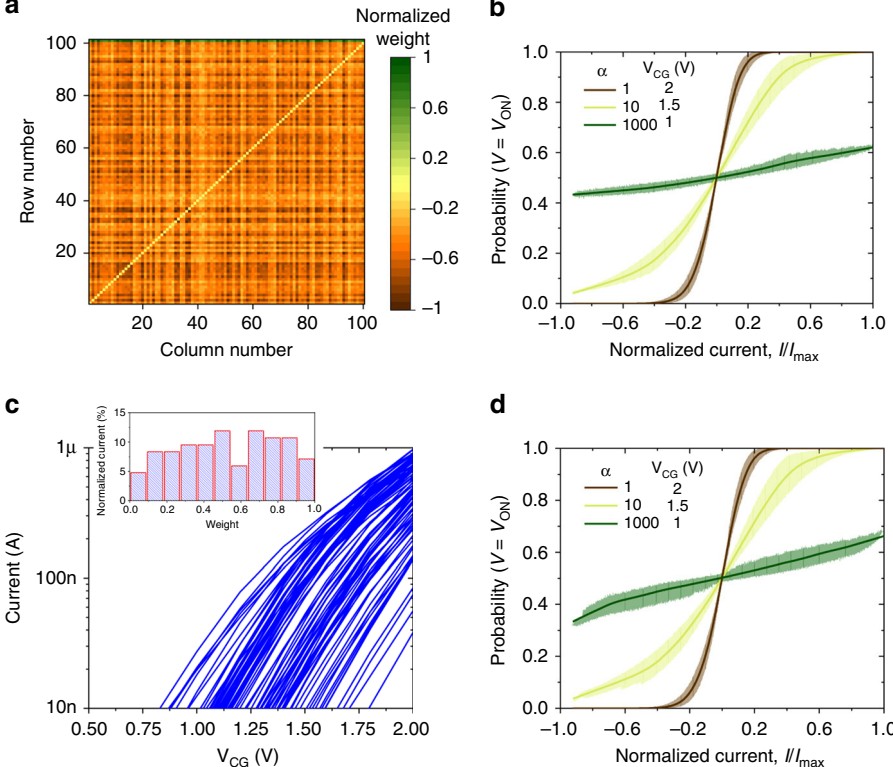

**Fig. 4 Non-idealities in flash-memory-based stochastic dot-product circuits. a** The considered distribution of neural network weights, normalized separately to its maximum value for positive (array) and negative (bias) weights. The bias weights are shown at the very top of the array. **b** Smearing of the stochastic neuron function due to input-dependent output-referred current noise in differential circuits for three computing temperatures. The thicker lines show the simulated values obtained by applying 10k randomly chosen inputs (neuron states) across all neurons, while solid lines show their averages. **c** Measured subthreshold slope $I$–$V$s for 100 memory cells. The inset shows the histogram of corresponding normalized synaptic weights, defined as $I_{cell}(V_{CG} = 2.0 \text{ V})/(I_{cell})_{max}$, when using $(I_{cell})_{max} = 973$ nA. **d** Simulated smearing of the stochastic neuron function due to the combined effect of subthreshold slope variations and differential summation in floating-gate memory implementation. The data were obtained similarly to that of panel (**b**), with only difference that subthreshold slopes for each device were randomly selected from the measured distribution and kept fixed during simulations of different inputs

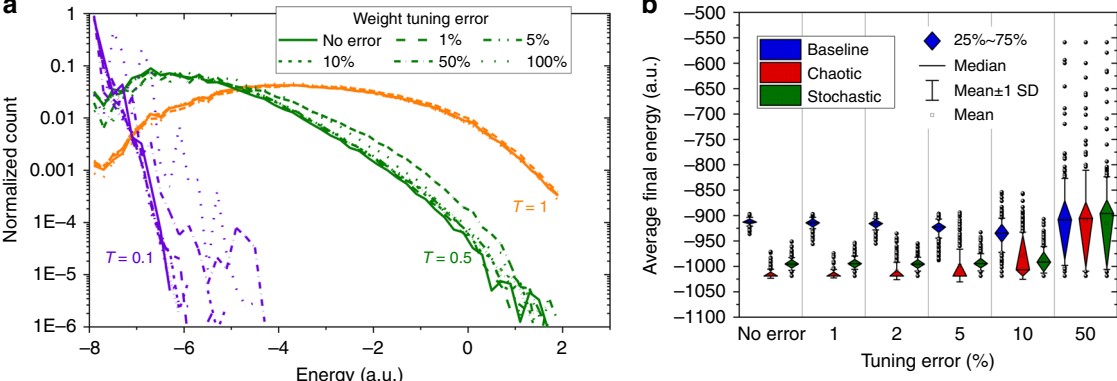

**Fig. 5** The impact of weight precision on functional performance. The simulation results for (**a**) energy distribution of RBM on Fig. 2a and (**b**) energies after 80th epoch of graph-partitioning problem of Fig. 3c, obtained with different assumptions for the conductance tuning. For RBM simulations, the data were collected over 500 epochs, and were averaged over 500 different trials. In neurooptimization experiment, 200 sets of weights were generated for each case of tuning error. A single data point on a graph represents an energy achieved after 80$^{th}$ epoch, averaged over 16 × 10 trials (10 runs for each of the 16 initial states) for a specific set of weights. For clarity, data point inside 25–75% are not shown. The tuning error was simulated by choosing randomly weights from the range of target value × [1−tuning error, 1 + tuning error]. To make the comparison meaningful, the energy is calculated assuming target (error-free) weights in both panels

memristor technology can be reduced by scaling-down device features[24]. Moreover, memristors with suitable range of resistances based on other materials have been also recently reported[66,67] and the further progress in this direction can be helped by development of foundry-compatible active metal-oxide memristor (1T-1R RRAM) macros[41,68].

Similar to other applications[53], a limited tuning precision and switching endurance for memristors and flash memories should be adequate for "inference"-like computations in both studied applications. For example, simulation results in Fig. 5 show almost no degradation in performance for up to ~5% and ~10% tuning errors (which is crudely equivalent to 3 and 2 bits of weight precision) for the studied RBM network and graph-partitioning problem, respectively. We also envision that the proposed neurooptimization hardware will be the most useful for computationally intensive problems, and thus require relatively infrequent weight re-tuning because of longer runtimes. In principle, implementations based on high-endurance digital memories, such as ferroelectric devices[69], would broaden the application space for the proposed circuits, e.g., enabling RBM training. Such implementations, however, would require multiple digital devices per synaptic weight, resulting in sparser designs with worse performance and energy efficiency.

In summary, we proposed to utilize extrinsic and intrinsic noise sources in mixed-signal memory-based circuits to implement efficient stochastic dot-product operation with runtime adjustable temperature. We then experimentally verified such idea by demonstrating memristive RBM and solving combinatorial optimization problem with floating-gate memory neuromorphic circuits. We believe that the future experimental work should focus on more promising continuous time/state networks with parallel state update[70] based on fully integrated hardware. The most urgent theoretical work includes modeling of the impact of the circuit and device non-idealities on the network functional performance, carrying out more rigorous comparison of annealing techniques for neurooptimization, as well as the development of larger-scale hardware suitable for more practical applications. In this context, it is worth mentioning that for the hardest combinatorial optimization problems, such as maximum clique problem, finding even largely suboptimal solution is challenging, which could greatly relax the device and circuit requirements.

## Methods

**Memristor array.** The RBM is implemented with a 20 × 20 passively integrated ("0T-1R") memristive crossbar circuits fabricated in UCSB's nanofabrication facility (Supplementary Fig. 5a–d). The active bilayer was deposited by low temperature reactive sputtering method, while crossbar electrodes were evaporated using oblique angle physical vapor deposition and patterned by lift-off technique using lithographical masks with 200-nm lines separated by 400-nm gaps. Crossbar electrodes are contacted to a thicker (Ni/Cr/Au 400 nm) metal line/bonding pad, which were implemented at the last step of the fabrication process.

Similar to ref. [24], majority of the devices were electroformed, one at a time, by applying one-time increasing amplitude voltage sweeps using automated setup. Automated "write-verify" tuning algorithm[62], involving alternative application of write and read pulses, was used for setting the memristor conductances to the desired values. Specifically, the memristors were formed/written one at a time using "V/2-biasing scheme", i.e. by applying half of the write voltages of the opposite polarity to the corresponding two lines connected to the device in question, while floating/grounding the remaining crossbar lines.

The formed memristors have fairly uniform switching characteristics, with set and reset voltages varying within 0.6–1.5 V and −0.6 to −1.7 V, respectively. The memristors' I–Vs are nonlinear at larger biases due to aluminum oxide tunnel barrier in the device stack, which helps with limiting leakage currents via half-selected devices during programming. Voltage drops across the crossbar lines are insignificant because of fairly large conductance of lines (~1 mS) compared to those of the crosspoint memristors (<36 μS). Supplementary Note 3 elaborates on the required further improvements in the device technology to avoid IR drop problem for more practical (i.e. larger-scale and higher-density) crossbar circuits.

More details on fabrication, electrical characterization, and memristor array operation can be found in refs. [24,33].

**Embedded NOR flash memory array.** The 12 × 10 arrays of floating-gate cells were fabricated in commercial 55-nm embedded NOR memory process, redesigned for analog applications (Supplementary Fig. 6a–c)[63]. (Such circuits were previously used to demonstrate vector-by-matrix multiplication with less than 3% weight/computing precision[63]). The array matrix is based on "supercells" (Supplementary Fig. 6a, b), which consist of two floating-gate transistors sharing the source (S) and the erase gate (EG) and controlled by different word (WG) and coupling (CG) gates. The cells are tuned using write-verify tuning procedure[62,63]. (Note that WG, D, and S supercell terminals are typically denoted by, respectively, WL, BL, and SL in the context of digital memory circuits. The new labels are more relevant to the considered application and were used to avoid possible confusion.)

After the weight tuning process had been completed, the network operation was performed using $V_D = 1$ V, $V_{WG} = 0.8$ V, $V_S = 0$ V, $V_{EG} = 0$ V, and $V_{CG} \in [1$ V, 2 V]. Such biasing conditions were mainly imposed by the requirement of keeping the transistors in the subthreshold region (Fig. 4c), ensuring large (>30 dB) dynamic range of SNRs, and minimizing the impact of subthreshold slope variations on weight scaling.

**Characterization setup.** The memristive crossbar circuit and flash memory chips were wire-bonded and mounted on custom printed circuit boards (Supplementary Figs. 5f and 6e). To steer the applied biases and sensed currents, the developed

board for flash memory chip also houses a microcontroller and a bank of low-leakage, low-noise ADG1438 analog multiplexers, while low-leakage Agilent E5250A switch matrix was used instead in memristor setup. The boards were connected to Agilent B1500A semiconductor device parameter analyzer, and an Agilent B1530A measurement unit to perform weight tuning, characterization, and all other measurements (Supplementary Figs. 5e and 6d). Agilent tools and printed circuit boards were controlled by C++ script running on a personal computer.

## Data availability
The data that support the plots within this paper and other findings of this study are available from the corresponding author upon reasonable request.

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

## Acknowledgements

This work was supported by Google Faculty award and a gift from the Institute for Energy Efficiency, UC Santa Barbara. The authors are thankful to F. Merrikh Bayat, X. Guo, and H. Nili for the background work on the flash-memory-based circuits and memristor characterization.

## Author contributions

M.M. and D.S. conceived the original concept. M.P. fabricated memristors. M.M. performed measurements and simulations. D.S. wrote the manuscript.

## Competing interests

The authors declare no competing interests.
