## [Peer Review File · Nature Communications]

Reviewers' comments:

Reviewer #1 (Remarks to the Author):

The present manuscript reports on neurocomputing using a combination of floating gate memories and memristive devices.

The authors discuss particular examples and provide an outlook for further development. The article is clearly structured, the results are original and the topic is suitable for the journal.

I do not have remarks on the technical approach and the data evaluation. However, I have some concerns regarding covering the broad interest of the readership of Nature Communications, that should be addressed prior to a decision on the manuscript.

Comments:

The authors indeed provide a nice implementation of two different types of memories to achieve improved functionalities. On the other hand, there is a large number of papers in the recent time publishing different approaches in reproducing, or achieving to different extent neuromorphic functionalities. It is at that point for me not immediately clear, what is the decisive advantage of this approach. I am a bit missing a comparative discussion highlighting this point, compared to other approaches.

Without underestimating the technical soundness of the experimental results, at that stage to me the manuscript is more suitable for focused journal. The authors should clearly differentiate their approach.

In that respect, I am also missing an interdisciplinary approach, involving also other scientific fields (e.g. materials science, physics etc.). The paper is strongly focused on data computation.

Reviewer #2 (Remarks to the Author):

This work presents the development of efficient specialized hardware for stochastic neural networks, particularly for Boltzmann machines. The functionality of the proposed approach is experimentally verified for two representative applications: (1) implementing neural network capable of solving a four-node graph-partitioning optimization problem, and (2) realizing restricted Boltzmann machine with 10 input and 8 hidden neurons.

Generally, this is an interesting work. I like that this work demonstrated two approaches to implement product-sum operation. This enables the revised manuscript has the position to compare the pros and cons between metal-oxide memristors and floating-gate memory approaches.

(1) Please discuss on the tolerance of the variation in memristor conductance for the dot-product operation (Figure 2 and Figure 3).

(2) Please specify the metal-oxide memristors cell structure employed in this work. Is it 1T1R? 1S1R? or pure memristor crossbar? If it is the pure-memristor crossbar structure, please add discussion on how the sneak current affects accuracy of reading selected row/column?

(3) For memristor-based circuits (Figure 1b), please add description in appendix on how to do the

write operation for storing the weight? Any design consideration should be highlighted?

(4) For floating-gate memory approach, as shown in Figure S3(c), please add discussion on how (principal or methodology) to choose the voltage of CG to get the required current for dot product?

(5) Usually, the loading of PCB (connection from chip to ADC) is not negligible. Please specify that the overhead of PCB is this also taken into consideration in the measured results?

(6) Fig.3(a) does not match with Fig. S5(b)/(c). The product-sum behavior of floating-gate, stack-gate, and 1T1R devices are different. If this work is based on floating-gate, please use consistent structure in figures.

(7) Please add discussion on how to apply proposed two schemes to a larger network. Readers would be highly interested in this topic.

(8) I like that this work demonstrated two approaches to implement product-sum operation. This work has the position to discuss and compare the pro and con between memristor and floating-gate memory approaches. I believe many readers would like to see this discussion and know which device is better for product-sum operation.

(9) There are three silicon-verified ReRAM based product-sum full-macro works published in ISSCC in recent two years (2018-2019). Though those works might have different applications from this work. To meet the high standard of Nature publications, the revised manuscript may strengthen the reference list by adding those works. It would be better to briefly discuss the difference or advance of proposed approach over those ISSCC works.

(10) The bit-precision of input, weight, and output precision are not clearly specified. Please explain the bit-precision of input, weight and output of the memristor and flash cells used in this work. What is the output precisions? Is there any potential impact of the output precision on the stochastic neuron behaviors?

(11) For revised manuscript, it would be better to give detailed schematics for the circuit implementation used in this work.

(12) Minor issues

a. many symbols used in the figures lack of explanation. (e.g. the input/output drivers with single or double pins in Fig.1d, the triangles with "x", etc...). Please clearly explain all symbols used in the figures.

Reviewer #3 (Remarks to the Author):

The work 'Versatile Stochastic Dot-Product Circuits Based on Nonvolatile Memories for High Performance Neurocomputing and Neural Optimization' by M. R. Mahmoodi et al. presents a study of restricted Boltzmann machine (RBM) in memory arrays in presence of external noise. The concepts of RBM, neuro-optimization, simulated annealing, and other computational concepts adopted in the manuscript, are all well known since the seminal work by Hopfield and other pioneers. Solving similar problems in the memory, e.g. by using crosspoint arrays of memory devices such as memristors and nanomagnets have also been presented [42,48,50]. The concept of dot-product engine consisting of an analogue or digital memory array where the product is conducted physically by multiplication of the applied voltage with the linear device conductance by Ohm's law is also well established in the

literature [25,27,30]. The topic is timely and relevant both scientifically and technologically as also confirmed by current research efforts by other groups (arXiv: 1903.11194).

A main challenge in these types of neural networks is exploiting 'physical' noise directly from the memory devices, or from hardware in general, instead of adopting pseudo-random noise generated from algorithms. Memristors and other memory devices have been in fact proposed as noise and random number generators in recent works, by taking advantage of the relatively large stochastic fluctuations of physical device phenomena, e.g., ionic migration and thermal noise. Given this underlying motivation, I do not think this work really makes significant steps to show how to extract and manipulate noise from device physics, which should be given the main emphasis. As a result, despite the work is well organized, sufficiently clear and rich of extensive references, I am not convinced there is sufficient groundbreaking novelty in this work to justify publication in Nature Communication at this stage. I am summarizing the main comments in the following.

- In both the memristive and the flash implementation of the RBMs, the noise is added externally, presumably by means of a software generator of noise and analogue addition to the applied voltage. This is a main limitation to demonstrate the feasibility of the concept, therefore experiments should include internal, physical noise from devices to support the memory-based RBM.

- Thermal noise is assumed for memristors and shot noise is assumed for Flash memory based on ballistic regime of channel transport in the device. I do not think this is accurate, as memristor noise is known to obey $1/f$ or $1/f^2$ (random telegraph noise) dependence on frequency. Similarly, I do not think that 55nm NOR flash are ballistic, therefore noise is most probably of $1/f$ type. Please provide experimental noise for the devices under study as a function of frequency to support your assumption.

- The neuron implementation is not clear. Please specify the hardware structure of the neuron, e.g., leaky integrate and fire, and the corresponding input/output characteristics.

- The graph partitioning problem is addressed only by the Flash implementation of the RBM, while the memristive network is used only to generate the sigmoidal function and energy distribution at various T . Please show hardware solution of the same problem, e.g., a relatively small graph or optimization problem, to allow comparison between the Flash and memristor hardware. Otherwise, the two parts of the manuscript are rather separate and hard to compare.

Detailed responses to reviewer’s comments

The authors would like to thank all reviewers for useful comments and suggestions. In response, we have revised the manuscript thoroughly, and believe that these changes have improved our paper substantially.

This document consists of 2 parts:

- (a) a brief list of the most significant changes and the new material in the manuscript, and
- (b) detailed responses to all referee’s comments, which also document all substantive changes made in the text, figures and figure captions of the paper.

(a) Significant changes, new material, and corrected typos in the revised version

- New Table S2, which outlines recently suggested bio-inspired hardware approaches for solving combinatorial optimization problems (in response to Referee #1)
- New Figure S4, showing simulation results for the sensitivity of studied Boltzmann Machine applications to weight precision (in response to Referee #2)
- New Figure S3b showing experimental results for memristor-based neurooptimization (in response to Referee #3)

(b) Detailed responses to reviewer’s comments

For convenience, the original comments/suggestions made by the Referees are numbered and typeset in blue, our responses are provided in black, while the yellow background highlights the changes in the revised version. Some of the similar comments are grouped together to avoid redundancy in responses.

Reviewer #1:

- 1) The present manuscript reports on neurocomputing using a combination of floating gate memories and memristive devices.

The authors discuss particular examples and provide an outlook for further development. The article is clearly structured, the results are original and the topic is suitable for the journal. I do not have remarks on the technical approach and the data evaluation...

The authors indeed provide a nice implementation of two different types of memories to achieve improved functionalities...

Without underestimating the technical soundness of the experimental results ...

We would like to thank the reviewer for this assessment.

- 2) ...However, I have some concerns regarding covering the broad interest of the readership of *Nature Communications*, that should be addressed prior to a decision on the manuscript.

Given the interdisciplinary scope of the considered work and potentially high appeal for the general audience we believe that our work fits *Nature Communications* better than a more focused and adapted journal. For example, a paper published in more focused *Nature Electronics* may not be immediately noticed by the neural network, machine learning, and computer science communities. Also, the popularity of neurooptimization hardware, one of the focuses of our work, can be further indirectly evidenced by many publications on similar topic in interdisciplinary journals like *Nature*, *Nature Communications*, *Nature Scientific Reports*, *Science* – please see new Table S2 added in the revised Supplementary Information.

- 3) ...On the other hand, there is a large number of papers in the recent time publishing different approaches in reproducing, or achieving to different extent neuromorphic functionalities. It is at that point for me not immediately clear, what is the decisive advantage of this approach. I am a bit missing a comparative discussion highlighting this point, compared to other approaches.

... at that stage to me the manuscript is more suitable for focused journal. The authors should clearly differentiate their approach.

It seems that Referee is asking to compare the proposed approach to previously suggested neuromorphic hardware. We believe that providing a high quality discussion on such a broad request would require comparing different neural network models and applications and hence more suitable for a lengthy review paper. Please note that our introduction is already rather long – ~3.5 pages out of ~11.5 pages total of text. Replacing the existing text in the introduction does not seem like a good option too as it may create other problems.

Nevertheless, to address the broad nature of this request, a very brief general motivation on the studied type of neuromorphic hardware and comparison to other common bio-inspired approaches are provided below in Figure R1 and its caption.

On the other hand, we agree that the paper would benefit from a more-focused comparison of previously suggested neuromorphic hardware, that is similar to our work. Such comparison is added to the revised Supplementary Information as a Table S2, which highlights the utilized device technology, considered applications, and the type of the demonstration.

- 4) In that respect, I am also missing an interdisciplinary approach, involving also other scientific fields (e.g. materials science, physics etc.). The paper is strongly focused on data computation.

We respectfully disagree with this comment. We believe that our work involved novel/nontrivial approaches in several disciplines. For example, the success of our work relied on a proper exploitation of device physics, in particular utilization of noise in memory devices and circuits for stochastic operation, utilization of non-mainstream optimization algorithms and mapping such algorithms to the proposed hardware, fabrication of novel circuits based on emerging memory devices and demonstration of their successful operation.

[Redacted]

Figure R1. Neuromorphic computing models. The figure shows major types of artificial neural networks and attempts to classify such networks by their resemblance to biology and complexity of neural network components. Only most common types are shown – for example, not listed neural networks with advanced features, like short-term potentiation in synapses, would be located even further up and right on the figure. The resemblance to the biology is represented by the number of borrowed features, with more features added when moving up along y axis. The general motivation for increasing biological plausibility is the belief that it will help with building systems matching human brain performance. The richer functionality is characteristic of higher complexity of neuromorphic hardware and/or more challenges in building such networks in hardware. Two important points are also highlighted in this figure. First is that though a more complex networks, like spiking neural networks, are very promising, they are generally not practically useful today and in need of certain algorithmic breakthrough. On the other hand, feed-forward and recurrent networks, with different types of connectivity and topologies (e.g. deep convolutional) are now commercially used for many applications. Probabilistic neural networks based on Boltzmann machines are also started being used commercially, e.g. in recommendation systems, albeit to lesser extent for now but their applications are rapidly growing. The second important point is that all neural networks rely heavily on vector-by-matrix multiplication operation. In addition to that, more complex neural networks require other types of functionalities, e.g. stochastic neurons in Boltzmann machines, spike-timing-dependent-plasticity in spiking neural networks, etc.

Reviewer #2:

- 5) This work presents the development of efficient specialized hardware for stochastic neural networks, particularly for Boltzmann machines. The functionality of the proposed approach is experimentally verified for two representative applications: (1) implementing neural network capable of solving a four-node graph-partitioning optimization problem, and (2) realizing restricted Boltzmann machine with 10 input and 8 hidden neurons. Generally, this is an interesting work. I like that this work demonstrated two approaches to implement product-sum operation. This enable the revised manuscript has the position to compare the pros and cons between metal-oxide memristors and floating-gate memory approaches.
... I like that this work demonstrated two approaches to implement product-sum operation.

This is accurate description of our work and we would like to thank reviewer for such positive assessment.

- 6) Please discuss on the tolerance of the variation in memristor conductance for the dot-product operation (Figure 2 and Figure 3).

To address this comment we have simulated how functional performance is changing with weight precision both applications studied in our paper. These preliminary results are now added in Supplementary Information (see new Figure S5 its discussion in Section S5) and also briefly mentioned in the main text (p. 12, 3rd paragraph – also shown below for convenience).

“It is worth noting that similar to other applications [53], a limited tuning precision and switching endurance for memristors and flash memories should be adequate for “inference”-like computations in both studied applications. For example, simulation results in Fig. S4 show almost no degradation in performance for up to ~5% and ~10% tuning errors (which is crudely equivalent to 3 and 2 bits of weight precision) for the considered RBM network and larger-scale graph partitioning problem, respectively.”

- 7) Please specify the metal-oxide memristors cell structure employed in this work. Is it 1T1R? 1S1R? or pure memristor crossbar? If it is the pure-memristor crossbar structure, please add discussion on how the sneak current affect accuracy of reading selected row/column?

We believe that “0T1R” qualifier would be more accurate description of the fabricated memristor circuits. Such definition implies “passively” integrated memristors, i.e. memory array in which each memory cell consists of only resistive switching element formed at the cross-section of the crossbar lines - see, e.g. top-view of such crossbar circuit in Figure 2b. The static I - V s of memristors are nonlinear at larger voltages – please take a look at figure S1b in Ref. 25. Similar to other access devices (“1T” or “1S”), such nonlinearity helps with decreasing leakage currents via half-selected devices during programming, though the change in conductance is not as sharp as in the typical “1S” device of “1S1R” cells.

The sneak path currents are either non-existent (for inference operation) or negligible (for read operation during conductance tuning), because of the used biasing scheme in which all lines are tied to some voltages and large electrode line conductance (~1 mS) compared to those of the cross-point memristors (which have largest conductance of 36 μ S) - please see Supplementary Note 1 in Ref. 25.

The fact that the fabricated crossbar circuits are passive is mentioned in few places in the paper. To further address this comment we have modified the original text and added new sentences as following:

“The restricted Boltzmann machine is implemented with a 20×20 passively integrated (“0T-1R”) memristive crossbar circuits fabricated in UCSB’s nanofabrication facility (Fig. S6a-d)...”

“... The memristors’ I - V s are nonlinear at larger biases due to aluminum oxide tunnel barrier in the device stack, which helps with limiting leakage currents via half-selected devices during programming. The sneak path currents during network operation and read phase of tuning algorithm are negligible, because all lines are always tied to some voltages and also very large conductance of lines (~ 0.5 mS) compared to those of the cross-point memristors (< 36 μ S), allowing for $< \sim 5\%$ computing error.”

More details on fabrication, electrical characterization, and memristor array operation can be found in Refs. 25 and 34.”

We have also found and corrected a bug, which might have been related to this comment. The following sentence

“In our first experiment, we focused on the demonstration of a restricted Boltzmann machine using 20×20 crossbar circuits with passively integrated Pt/Al/TiO_{2-x}/Pt memristors (Fig. 2).”

on page 7 is now corrected to

“In our first experiment, we focused on the demonstration of a restricted Boltzmann machine using 20×20 crossbar circuits with passively integrated Pt/Al₂O₃/TiO_{2-x}/Pt memristors (Fig. 2).”

8) For memristor-based circuits (Figure 1b), please add description in appendix on how to do the write operation for storing the weight? Any design consideration should be highlighted?

Our write procedure is very common. The devices are programmed one by one using $V/2$ procedure, i.e. by biasing two lines connected to the device with $-V/2$ and $+V/2$ while grounding other lines.

To address this comment we have added the following paragraph in the Methods section:

“Similar to Ref. 25, majority of the devices were electroformed, one at a time, by applying one-time increasing amplitude voltage sweeps using automated setup. Automated “write-verify” tuning algorithm [62], involving alternative application of write and read pulses, was used for setting the memristor conductances to the desired values. Specifically, the memristors were formed/written one at a time using “ $V/2$ -biasing scheme”, i.e. by applying half of the write voltages of the opposite polarity to the corresponding two lines connected to the device in question, while floating/grounding the remaining crossbar lines.”

For clarity, we also moved one sentence from Method section and added another one to the main text as follows:

“... The integrated memristors are sufficiently uniform for programming with less than 5% tuning error, and have negligible conductance drift over time...”

... (More details on the memristor technology and crossbar circuit operation is provided in Methods section.)”

- 9) For floating-gate memory approach, as shown in Figure S3(c), please add discussion on how (principal or methodology) to choice the voltage of CG to get the required current for dot product?

We have added the following discussion to Methods sections:

“After the weight tuning process had been completed, the network operation was performed using $V_{BL}=1$ V, $V_{WL}=0.8$ V, $V_{SL}=0$ V, $V_{EG}=0$ V, and $V_G \in [1$ V, 2V]. Such biasing conditions were mainly imposed by the requirement of keeping the transistors in subthreshold region, ensuring large (>30 dB) dynamic range of signal-to-noise ratios, and minimizing the impact of subthreshold slope variations on weight scaling – see Section S5 of the Supplementary Information.”

- 10) Usually, the loading of PCB (connection from chip to ADC) is not negligible. Please specify that the overhead of PCB is this also taken into consideration in the measured results?

We certainly agree that this could be a problem for high speed measurements. However, due to rather moderate sampling frequency operation - 1 MHz for memristor and 10 KHz for Flash memory experiments - this was not an issue for the experiment performed in our paper. Also, please note that in both experiments the leakages via muxes in custom-made PCB and commercial switching matrix were limited to below < 300 pA.

- 11) Fig.3(a) does not match with Fig. S5(b)/(c). The product-sum behavior of floating-gate, stack-gate, and 1T1R devices are different. If this work is based on floating-gate, please use consistent structure in figures.

Thank you for catching this mistake. We have revised Figure 3a by using the proper device schematics for floating gate transistor and adding more details. The Figure 3a is now consistent with Fig. S7b (which was Fig. S5 in the original manuscript).

- 12) Please add discussion on how to apply proposed two schemes to a larger network. Readers would be highly interested in this topic.

We believe that it is very non-trivial issue, which will require comprehensive and synergistic development of algorithms and hardware. Hence, we have extended the future work in summary of the revised manuscript as following:

“The most urgent theoretical work includes ... as well as the development of larger-scale hardware suitable for more practical applications.”

Please note that we don’t see any obvious showstoppers for such development. Our confidence is rooted in very encouraging results for practically-useful neuromorphic inference accelerators based on similar to the considered in our work memory technologies.

- 13) This work has the position to discuss and compare the pro and con between memristor and floating-gate memory approaches. I believe many readers would like to see this discussion and know which device is better for product-sum operation.

We completely agree with this comment. ~0.75-page-long discussion on the comparison of two considered approaches is provided in the second and third paragraph of the Discussion section.

- 14) There are three silicon-verified ReRAM based product-sum full-macro works published in ISSCC in recent two years (2018-2019). Though those works might have different applications from this work. To meet the high standard of Nature publications, the revised manuscript may strength the reference list by adding those works. It would be better to briefly discuss the difference or advance of proposed approached over those ISSCC works.

We are now citing two papers from ISSCC’18 and ISSCC’19 conference related to RRAM macros and added the following text to the revised version:

“... and the further progress in this direction can be helped by development of foundry-compatible active metal-oxide memristor (1T-1R RRAM) macros [12, 66].”

Please note that the main difference is that in our work we use more prospective (but also more challenging) passive (0T1R) technology.

- 15) The bit-precision of input, weight, and output precision are not clearly specified. Please explain the bit-precision of input, weight and output of the memristor and flash cells used in this work. What is the output precisions? Is there any potential impact of the output precision on the stochastic neuron behaviors?

The approximate input/weight/output effective precisions for eFlash and memristor experiments are, respectively, 1 bit / 4 bit / 4 bit and 1 bit / 3.5 bit / 3.5 bit.

Specifically, all studied networks are based on binary neurons so that the inputs are always 1 bit. In our application demonstrations, floating gate memory cells’ and memristors’ currents are set, respectively, with 3% and 5% tuning error at operating (or final) biasing conditions, which is crudely equivalent to 4 and 3.5 bits of weight precision. (The precision could be worse initially for the experiments with dynamic weight scaling using eFlash memory circuits - this issue is discussed in detail in Section S5 of the Supplementary Information.) Finally, our prior work on the deterministic vector-by-matrix multipliers using similar flash memory (memristor) circuits showed that compute error is less than 3% (5%) - please see Ref. 63 (Fig. S8 and its discussion in Ref. 25).

To address this comment we have added in Method section “... allowing for < ~5% computing error.” and “... less than 3% weight/computing precision”

We believe that the impact of weight / computing precision and other non-idealities is not a trivial issues and require further thorough investigation. We mention this point in a summary. It is worth mentioning, that similarly to previously observed results for stochastic synapses, stochastic neuron operation would likely relax the requirement for computing precision – see, e.g. the discussion of this point in the second paragraph of the manuscript.

- 16) For revised manuscript, it would be better to give detailed schematics for the circuit implementation used in this work.

To address this comment, we have redrawn Figure 3a of the manuscript and added new Fig. S2. We believe that these figures, together with Figs. S6 and S7 provide all details of the experiment.

- 17) Minor issues: a. many symbols used in the figures lack of explanation. (e.g. the input/output drivers with single or double pins in Fig.1d, the triangles with “x”, etc...). Please clearly explain all symbols used in the figures.

We have added more information to / updated Figs. 1 and 3 and their captions in the revised manuscript.

Reviewer #3:

- 18) The topic is timely and relevant both scientifically and technologically...
... the work is well organized, sufficiently clear and rich of extensive references...

We would like to thank Referee for this assessment of our work.

- 19) The work ‘Versatile Stochastic Dot-Product Circuits Based on Nonvolatile Memories for High Performance Neurocomputing and Neural Optimization’ by M. R. Mahmoodi et al. presents a study of restricted Boltzmann machine (RBM) in memory arrays in presence of external noise. The concepts of RBM, neuro-optimization, simulated annealing, and other computational concepts adopted in the manuscript, are all well known since the seminal work by Hopfield and other pioneers. Solving similar problems in the memory, e.g. by using crosspoint arrays of memory devices such as memristors and nanomagnets have also been presented [42,48,50]. The concept of dot-product engine consisting of an analogue or digital memory array where the product is conducted physically by multiplication of the applied voltage with the linear device conductance by Ohm’s law is also well established in the literature [25,27,30]...

...Memristors and other memory devices have been in fact proposed as noise and random number generators in recent works, by taking advantage of the relatively large stochastic fluctuations of physical device phenomena, e.g., ionic migration and thermal noise.

We completely agree with this discussion and believe these points are clearly mentioned in the abstract and introduction of our manuscript. Please note that one of the major novelty of our work is in taking advantage of *both* stochastic properties of devices and very efficient dot-product memristive circuits, and experimentally demonstrating it. To the best of our knowledge, this was never reported before.

Please also note that Refs. 42 and 48 present simulation results of the network based on experimental measurements from single devices, while Ref. 50 is showing experimental results

based on smaller 3×10 array, and as Referee correctly pointed out, without relying on hardware generated noise.

20) Given this underlying motivation, I do not think this work really makes significant steps to show how to extract and manipulate noise from device physics, which should be given the main emphasis.

.... I am not convinced there is sufficient groundbreaking novelty in this work to justify publication in Nature Communication at this stage. I am summarizing the main comments in the following.

... A main challenge in these types of neural networks is exploiting ‘physical’ noise directly from the memory devices, or from hardware in general, instead of adopting pseudo-random noise generated from algorithms...

... In both the memristive and the flash implementation of the RBMs, the noise is added externally, presumably by means of a software generator of noise and analogue addition to the applied voltage. This is a main limitation to demonstrate the feasibility of the concept, therefore experiments should include internal, physical noise from devices to support the memory-based RBM.

We agree with Referee that exploiting physical noise from hardware is more challenging than adopting pseudo-generated noise from algorithms (which is, e.g., how it was done in the arXive paper mentioned by Referee). This is why we believe it is a major strength of our paper - all experimental results reported in Figs. 2 and 3 were obtained by injecting the noise, with characteristics show in insets of Fig. 2c and 3b, directly from the hardware read-out circuitry, without relying on any post-processing in software.

We apologize for confusion. To address this issue we have added/modified the text in several places of the revised manuscript:

“We experimentally verify stochastic dot-product circuits based on metal-oxide memristors and embedded floating-gate memories by implementing and testing representative Boltzmann machine networks with non-binary weights and hardware-injected noise.”

“...Figure 2c shows stochastic dot-product results when utilizing external white noise with a fixed standard deviation (inset of Fig. 2c), which was injected directly in the hardware from the readout-circuitry, which was injected directly in the hardware from the readout-circuitry...”

In our main RMB experiment, we first apply randomly generated digital voltages to the vertical crossbar lines connected to visible neurons, then sample output currents on the horizontal crossbar lines feeding hidden neurons, and convert sampled values to the new digital voltages of hidden neurons according to the signs of the corresponding differential currents. Note that only functionality of a sensing circuit and latch (i.e. applying step function to the sensed currents and holding the resulting digital value) are realized in a software, while the probability function of the Eq. 3 is implemented directly in the hardware. In the next step, the calculated voltages at the hidden neurons are applied to horizontal lines and the new voltages at the input neurons are computed similarly to the forward pass. The voltages at the input and hidden neurons represents

the new state of the network after one forward/backward state update (“epoch”) and are used to calculate its energy according to the Eq. 1. These updates are repeated multiple times in a single run of the experiment.”

“Similar to RBM study, fixed white noise was added externally directly from the readout circuit (inset of Fig. 3b), while other peripheral functions were emulated in the software.”

We also modified Fig. 3a and its caption to further clarify this point.

- 21) Thermal noise is assumed for memristors and shot noise is assumed for Flash memory based on ballistic regime of channel transport in the device. I do not think this is accurate, as memristor noise is known to obey $1/f$ or $1/f^2$ (random telegraph noise) dependence on frequency. similarly, I do not think that 55nm NOR flash are ballistic, therefore noise is most probably of $1/f$ type. Please provide experimental noise for the devices under study as a function of frequency to support your assumption.

We agree that the noise power is higher for ($1/f$ or $1/f^2$) pink noise at low ($< \sim 1$ MHz) frequencies for both approaches. However, the low frequency operation is not representative for the most practical version/use of the proposed circuits in which memory arrays are tightly integrated with CMOS-based neurons. According to the prior work on deterministic VMMs implemented with embedded NOR flash [53, 54] and memristors [25], both types of *integrated* circuits could readily operate at > 100 MHz frequencies at which thermal/shot noise would be by far dominating.

To address this comment, we have clarified in the revision that the presented analysis is intended for the most practical, fully-integrated version of the proposed circuits. We have also clarified that the shot noise would be more representative of nanoscale MOSFETs with channel length below ~ 10 nm, while both thermal and shot noise should be considered in 55-nm MOSFETs. In particular, the following changes were made:

“To analyze stochastic operation, let us consider normally distributed independent noise sources. This assumption is justified due to the dominant white (thermal and/or shot) intrinsic noise for the most practical >100 MHz bandwidth operation, which would be realistic for both floating -gate transistor and memristor-based analog circuits in which memory arrays are tightly integrated with peripheral circuits [53-54].”

“On the other hand, the intrinsic shot noise is characteristic of a ballistic transport in nanoscale floating-gate transistors with sub-10-nm channels [60,61].”

Please note that low frequency pink noise for similar technology memristors was already reported by our group in Fig. 3 of M. Prezioso et al., Proc. IEDM'15, pp. 17.4.1 – 17.4.4, while previous studies - see, e.g., Hideto Hidaka ed., “Embedded Flash Memory for Embedded Systems: Technology, Design for Sub-systems, and Innovations, 2018 - showed that current fluctuations in floating gate transistors are similar to those of the conventional MOSFETs. Also, detailed noise measurements at high frequencies are not possible with our current experimental setup, in part because we cannot remove the noise injected by the readout circuits and also because our laboratory is not currently equipped with low-noise characterization tools.

- 22) The neuron implementation is not clear. Please specify the hardware structure of the neuron, e.g., leaky integrate and fire, and the corresponding input/output characteristics.

Please note that the neurons are emulated in a software (which was explicitly stated in text). Still, to address this comment we have added a possible implementation in Figure S2.

23) The graph partitioning problem is addressed only by the Flash implementation of the RBM, while the memristive network is used only to generate the sigmoidal function and energy distribution at various T . Please show hardware solution of the same problem, e.g., a relatively small graph or optimization problem, to allow comparison between the Flash and memristor hardware. Otherwise, the two parts of the manuscript are rather separate and hard to compare.

To address this comment, we have performed additional neuro-optimization experiments based on memristor hardware. The results of these experiments are now added as new Figure 2b in the revised supplementary information. Please note that floating-gate memory circuits are less suitable for RBM networks in which information has to be propagated in both forward (from input to hidden neurons) and backward (from hidden to input neurons). We mentioned this issue in the Discussion section. We believe that given that memristor and flash memory technologies are representative of two major types of mixed-signal circuits, the inclusion of both approaches and discussion of their cons and pros add a value to the paper. (This point seems to be also appreciated by the Referee #2.)

1) ... as also confirmed by current research efforts by other groups (arXiv:1903.11194).

We are aware of that ArXive paper. Though impressive, this paper still has to go through rigorous review process and we decided not to cite it because of possible flaws. We can note now, however, that unlike our work that used experimentally injected noise, the noise in that arXive paper was simulated in software and, additionally, all experiments were performed with simpler binary-weight networks.

Reviewers' comments:

Reviewer #1 (Remarks to the Author):

The authors have responded to comments and the manuscript has been definitely improved. To me it is still not very immediate how the general readers will distinguish the particular results from the large number of papers on same topic. Also the paper is focused mainly on computing and I still do not see some physics behind. However, the strengths of the computing study are strong enough and I can recommend the manuscript for publication.

Reviewer #2 (Remarks to the Author):

The revised manuscript has addressed most of my concerns. Two minor issues need to be further clarified.

(1) In revised manuscript, the authors have added "The sneak path currents during network operation and read phase of tuning algorithm are negligible, because all lines are always tied to some voltages and also very large conductance of lines (~ 0.5 mS) compared to those of the cross-point memristors (< 36 μ S)..."

We respectfully disagree with this point. Although the sneak current can be possibly neglected in the crossbar array with relatively small size (e.g. the 20x20 "0T1R" crossbar array used in this work), its impact can become prominent with increasing memory capacity. As the capacity and density of the memory increases, the conductance of lines decreases monotonically because of their increasing lengths and limited widths. Besides, the current loading of WL/BLs in the "0T1R" crossbar also increase with increasing memory capacity. In order to perform network operations in memristive memory with large capacities, sneak paths in the memory array should be addressed properly.

(2) In supplementary figure 2, the authors presented a possible implementation of peripheral circuits. The authors seem use small triangles to represent multiplexers (MUXs). If MUXs are used here, it is recommended to use the standard symbols to represent them.

Reviewer #3 (Remarks to the Author):

I'd like to thank the authors for their reply and revision. One thing is still missing and unclear though. In their reply, the authors report that the 'experimental results reported in Figs. 2 and 3 were obtained by injecting the noise ... directly from the hardware read-out circuitry, without relying on any post-processing in software'. In another point of the manuscript, the authors state that 'detailed noise measurements at high frequencies are not possible with our current experimental setup, in part because we cannot remove the noise injected by the readout circuits and also because our laboratory is not currently equipped with low-noise characterization tools.'

So I am still confused about what type of hardware noise was actually injected in the experiment. Is the hardware noise obtained from memristors and Flash memories? if yes, at what frequency range? If not, where was the noise collected? All these aspects should be clearly reported in the manuscript, given that the use of hardware noise is a 'major strength of our paper' according to the authors themselves. I think that it might be a major strength only if the noise in the experiment is really collected from the memristor/Flash devices as in the underlying concept of the stochastic dot-product. Otherwise, I remain skeptical about the real progress introduced by the work.

List of Changes in Response to the Second Round of Reviewers’ Comments

The authors would like to thank the reviewers for reading the revised version of the manuscript. We have addressed the remaining comments as follows:

In response to comments by Referee #2 (reproduced below in blue):

1) In revised manuscript, the authors have added “The sneak path currents during network operation and read phase of tuning algorithm are negligible, because all lines are always tied to some voltages and also very large conductance of lines (~ 0.5 mS) compared to those of the cross-point memristors (< 36 μ S)...”

Although the sneak current can be possibly neglected in the crossbar array with relatively small size (e.g. the 20x20 “0T1R” crossbar array used in this work), its impact can become prominent with increasing memory capacity. As the capacity and density of the memory increases, the conductance of lines decreases monotonically because of their increasing lengths and limited widths. Besides, the current loading of WL/BLs in the “0T1R” crossbar also increase with increasing memory capacity. In order to perform network operations in memristive memory with large capacities, sneak paths in the memory array should be addressed properly.

Our response: We agree that in general sneak-path currents is very important issue for more practical (larger-scale and higher-density) crossbar circuits. Perhaps, there is just minor disagreement in terminology in that we consider IR drops and sneak-path currents as two separate problems. The IR drop problem can be loosely defined as having non-negligible voltage drop across crossbar lines, while sneak path current problem as having non-negligible currents via half-selected and/or un-selected crosspoint devices. The sneak path currents do result in larger IR drops. However, the most problematic IR drops in our circuits occur without any sneak path currents, and because of that the sneak path currents do not have much impact on the functionality of the studied application, even for the most prospective (scaled-up, higher-density) crossbar circuits.

To further clarify this point, Figure S8 shows sneak currents for the three distinctive operations in the proposed ex-situ-trained neuromorphic circuits: (1) writing and (2) reading the state during conductance tuning, and (3) inference operation. This figure shows that, by design, sneak-currents never occur at inference operation. Moreover, our previous analysis (Supplementary Note 1 from Ref. 25) showed that IR drop problem is the most severe at inference operation and reducing IR drops to acceptable levels for inference operation would automatically results in acceptable IR drops (in this case, mostly due to sneak path currents) to correctly perform read and write operations of the tuning procedure. It is worth noting that the larger sneak currents would lead to higher power consumption for the write operation, but it is not critical for the considered applications because of infrequent weight re-tuning. In this regard, the ex-situ-trained neural networks are different from digital memories, another common application of memristors/ReRAMs, for which write energy is one of the most important metrics.

The IR problem can be addressed by increasing line conductance and/or decreasing currents of the crosspoint devices. To further address the Referee’s comments, we have

- added new Figure S8 and its discussion in Supplementary Information Section 8, providing more details on the sneak path current and IR drop problems, and
- in light of the newly added material, replaced the sentence, which Referee was not happy with, to

“Voltage drops across the crossbar lines are insignificant because of fairly large conductance of lines (~ 1 mS) compared to those of the cross-point memristors ($< 36 \mu\text{S}$). Supplementary Section 8 elaborates on the required further improvements in the device technology to avoid IR drop problem for more practical (i.e. larger-scale and higher-density) crossbar circuits.”

2) In supplementary figure 2, the authors presented a possible implementation of peripheral circuits. The authors seem use small triangles to represent multiplexers (MUXs). If MUXs are used here, it is recommended to use the standard symbols to represent them.

Our response: Thank you for pointing to this issue. We have modified Figure S2 with proper symbols that should be used for analog MUXes.

In response to comments by Referee #3 (reproduced below in blue):

3) One thing is still missing and unclear though. In their reply, the authors report that the ‘experimental results reported in Figs. 2 and 3 were obtained by injecting the noise ... directly from the hardware read-out circuitry, without relying on any post-processing in software’. In another point of the manuscript, the authors state that ‘detailed noise measurements at high frequencies are not possible with our current experimental setup, in part because we cannot remove the noise injected by the readout circuits and also because our laboratory is not currently equipped with low-noise characterization tools.’

Our response: Our point is that we cannot characterize noise from memory cells, because of much higher noise floor of our measurement setup (switching matrix and Agilent tools) at the frequency ranges of interest ($> \sim 10$ KHz). It is worth mentioning that simple estimates show that the noise power from memory cells in both experiments should be significantly lower than the experimentally measured injected noise power from the read-out circuitry. Note that we have also simulated neurooptimization task with stochastic annealing approach using foundry models at proper frequencies (~ 100 MHz) to show that the fully integrated circuits will work with only the intrinsic noise (Fig. S2a).

4) So I am still confused about what type of hardware noise was actually injected in the experiment. Is the hardware noise obtained from memristors and Flash memories? if yes, at what frequency range? If not, where was the noise collected? All these aspects should be clearly reported in the manuscript, given that the use of hardware noise is a ‘major strength of our paper’ according to the authors themselves.

Our response: The noise was *not* from memory cells, but rather injected by the read-out circuitry. The frequency ranges were 1 MHz and 10/20 KHz for memristors and flash memory probabilistic circuit experiments, respectively. Specifically, for the flash memory setup, the noise was mostly generated at the custom-made switching matrix, while it was contributed by the combination of custom PCB, switching matrix, and B1530 tool in memristor experiments. We believe that in both cases such noise fits the definition of external noise shown on the equivalent circuit in Figure 1a.

We have already mentioned in multiple places in the manuscript that the noise is externally injected from the read-out circuitry and explicitly mentioned frequency ranges (caption of Figure 2 for memristor and pages 9 and 10 for flash memory experiments). To further address this comment, we have added the measured spectra of the externally injected noise in both experiments – please see new Figure S9 and its discussion in Supplementary Information.

In light of the added new Figure S9, we have also modified the main text as follows:

Figure 2c shows stochastic dot-product results when utilizing external noise, which was injected directly in the hardware from the read-out circuitry. The noise spectrum is rather flat at higher ($> \sim 1$ KHz) frequencies (Supplementary Information Fig. S9a), which results in approximately fixed standard deviation of the noise (inset of Fig. 2c) for the studied 1 MHz bandwidth.

5) I think that it might be a major strength only if the noise in the experiment is really collected from the memristor/Flash devices as in the underlying concept of the stochastic dot-product. Otherwise, I remain skeptic about the real progress introduced by the work.

Our response: We would like to point out that this comment contradicts with the previous one made by Referee in the previous round of reviews:

“... A main challenge in these types of neural networks is exploiting ‘physical’ noise directly from the memory devices, **or from hardware in general**, instead of adopting pseudo-random noise generated from algorithms...”

Additionally, we believe that externally added noise (Fig. 1a) is still a viable alternative to the internal noise, e.g. when higher speed is desired, so we deliberately considered both intrinsic (from memory cells) and external noise in our study.

Other changes:

In addition to the changes listed above, we have proofread the manuscript again, and

- fixed a newly found typo in Fig. 3b legend of the main text. Namely, the dimensions in legend in the original version were specified as “mA” while the correct ones are “ μ A”; and
- to improve readability (mainly avoiding confusion of $V_{1'}$ with V_1), we have change notation for the ON and OFF state voltages from $V_{1'}$ to V_{ON} and from $V_{0'}$ to V_{OFF} in text, figures, and captions.

REVIEWERS' COMMENTS:

Reviewer #2 (Remarks to the Author):

I am satisfied with his revision.

Reviewer #3 (Remarks to the Author):

The authors have further revised the manuscript and clarified the origin of the noise in their concept. The work is suitable for publication in this form.